# Discovery and Prediction Study of the Dominant Pharmacological Action Organ of *Aconitum carmichaeli* Debeaux Using Multiple Bioinformatic Analyses

**DOI:** 10.3390/ijms251810219

**Published:** 2024-09-23

**Authors:** Musun Park, Eun-Hye Seo, Jin-Mu Yi, Seongwon Cha

**Affiliations:** 1Korean Medicine (KM) Data Division, Korea Institute of Oriental Medicine, Daejeon 34054, Republic of Korea; eunhyes@kiom.re.kr (E.-H.S.); scha@kiom.re.kr (S.C.); 2KM Convergence Research Division, Korea Institute of Oriental Medicine, Daejeon 34054, Republic of Korea; jmyi@kiom.re.kr

**Keywords:** *Aconitum carmichaeli* Debeaux, dominant pharmacological action organ, docking analysis, network pharmacology, transcriptome analysis

## Abstract

Herbs, such as *Aconitum carmichaeli* Debeaux (ACD), have long been used as therapies, but it is difficult to identify which organs of the human body are affected by the various compounds. In this study, we predicted the organ where the drug predominantly acts using bioinformatics and verified it using transcriptomics. We constructed a computer-aided brain system network (BSN) and intestinal system network (ISN). We predicted the action points of ACD using network pharmacology (NP) analysis and predicted the dockable proteins acting in the BSN and ISN using statistical-based docking analysis. The predicted results were verified using ACD-induced transcriptome analysis. The predicted results showed that both the NP and docking analyses predominantly acted on the BSN and showed better hit rates in the hub nodes. In addition, we confirmed through verification experiments that the SW1783 cell line had more than 10 times more differentially expressed genes than the HT29 cell line and that the dominant acting organ is the brain, using network dimension spanning analysis. In conclusion, we found that ACD preferentially acts in the brain rather than in the intestine, and this multi-bioinformatics-based approach is expected to be used in future studies of drug efficacy and side effects.

## 1. Introduction

*Aconitum carmichaeli* Debeaux (ACD) is one of the natural medicines that has been widely used since ancient times, and it is currently still widely used. The main compounds of ACD are aconitine-type diterpene alkaloids that act as cardiotonic agents [1,2,3] (Appendix A). In addition, ACD has recently been found to have new therapeutic effects such as relieving peripheral neuropathic pain [4,5], improving age-related muscle atrophy [6], and exhibiting antiviral activity [7]. Although ACD has been studied for its effects on several diseases and various conditions, to our knowledge, no study has been conducted for predicting the organs that are dominantly affected.

The organ scope of pharmacological action of herbal medicine is determined based on a unique theory called ‘meridian affinity’ [8]. ACD acts on the heart and spleen meridian affinity systems [9]. Traditionally, the heart system is used to treat the circulatory and nervous systems, while the spleen is used to treat the digestive system [8]. This can be reinterpreted as ACD acting on the nervous and digestive systems in addition to the circulatory system including the heart [10]. For example, ACD is used for abdominal pain and diarrhea that occur at dawn because it acts on complex symptoms of the nervous and digestive systems, such as irritable bowel syndrome with predominant diarrhea [11]. Treatment based on ‘meridian affinity’ has been empirically proposed and used for treating patients; however, it does not provide an accurate range of action of data-based drug therapy. Therefore, a method is necessary to address this issue.

Bioinformatic analysis methods, including network biology and drug-induced transcriptome analysis, can be used to overcome these limitations. Network biology models biological phenomena using networks and predicts physiological mechanisms using multiple biological targets [12]. Drug-induced transcriptome analysis identifies drug action mechanisms using the relative expression levels of transcripts in drug-treated tissues [13,14,15]. The analysis that identifies the drug action mechanism in multiple targets by integrating the two methods is commonly known as network pharmacology (NP) [16]. The NP method is a powerful method that can effectively present the scope of the organ in which the drug acts and its therapeutic effects.

In this study, we explored the dominant pharmacological actions and therapeutic effects of ACD using various bioinformatic analyses. A computer-based organ network was constructed to simulate the brain and digestive system. Using the results of the NP platform and large-scale molecular docking analysis, we identified which networks among the brain system network (BSN) and the intestinal system network (ISN) were preferentially affected by ACD. Finally, we verified the dominant pharmacological actions of ACD and predicted its therapeutic effects using drug-induced transcriptomic experiments.

## 2. Results

### 2.1. Organ Network Construction Results

The BSN and ISN were constructed to predict the dominant pharmacological organs in which ACD acts. The BSN consisted of 1040 protein nodes and 3938 edges (Appendix A, Appendix A), and the ISN consisted of 366 protein nodes and 777 edges (Appendix A, Appendix A).

### 2.2. NP Analysis Results

The NP analysis results showed that the ACD information provided by the BATMAN-TCM contained 56 compounds, 393 targets interacting with the compounds, and 1543 compound–target interactions (Appendix A). Mapping the 393 targets to organ networks predicted 7.69% in the BSN and 4.92% in the ISN (Figure 1, Table 1). When the analysis was restricted to the nodes with the top 20% degree centrality values that acted as hubs in each network, 17.62% of the proteins were predicted in the BSN (Table 1 and Appendix A hit count column), and only 6.76% of the proteins were predicted in the ISN (Table 2 and Appendix A hit count column). The permutation test results showed that the components of the ACD had a statistically significant effect on the hub of the BSN but no significant effect on the ISN (Table 2).

### 2.3. Molecular Docking Analysis

A database search for molecular docking identified 92 compounds in ACD (Appendix A). Among the 854 druggable proteins provided by the Human Protein Atlas (HPA) database (Appendix A), 157 proteins comprised the BSN (Appendix A; 165 proteins including fractions) and 54 proteins comprised the ISN (Appendix A). According to the results of the docking analysis, 69 proteins in the BSN (Appendix A) and 31 proteins in the ISN (Appendix A) were predicted to have more than 10 significant interactions (Table 1). In an analysis using only nodes with the top 20% centrality values, 13.33% of the proteins were predicted to directly interact with proteins belonging to BSN hubs (Appendix A column, Degree > 11, 28/210, 13.33%), and 13.51% of the proteins were predicted to directly interact with proteins belonging to ISN hubs (Appendix A column; Figure 2, Table 2). Although there was no significant difference between the BSN and ISN in the hub ratio analysis, Compound Spring Embedding (CoSE) analysis showed that the BSN was centrally concentrated with 65 proteins, except for 4, whereas the ISN was relatively spread out. This suggests that ACD interacts with proteins that play a central role in the BSN. In addition, the permutation test results showed that the BSN significantly interacted with hub proteins, whereas the ISN did not pass the significance level (Table 2). The docking analysis showed that core proteins predicted to interact with more than 50 compounds included in ACD were more abundant in the brain than in the intestine. In the brain, CHRM1, RPE65, CHRM4, GRIK4, CACNA1B, and CACNA1E were selected as core proteins, which were mainly related to the functions of neurotransmitters and calcium channels (Appendix A). In the intestine, only CHRM2 and CACNA1C were selected as core proteins, which were also related to neurotransmitters and calcium channels, similar to their action in the brain (Appendix A).

### 2.4. Transcriptome Verification Experiments

#### 2.4.1. Comparison of the Number of Differentially Expressed Genes (DEGs) by Cell Line

In this study, we used compounds from ACD collected from the database to predict whether they would act more specifically on either the brain or intestine. To verify whether these predictions held true in real-world situations, we performed validation experiments using ACD extracts. The number of DEG was compared in the transcriptome expression data obtained by treating SW1783 and HT29 cells with a high-dose water extract. Despite using the same process, the number of DEGs in SW1783 and HT29 at high doses showed a large difference. At high doses, 1388 transcripts were upregulated and 1465 transcripts were downregulated in SW1783 cells; however, in HT29 cells, only 137 transcripts were upregulated and 105 transcripts were downregulated, showing a difference of approximately 10 times (Figure 3A,B). In the organ network, 16.54% of SW1783 DEGs were hits in the BSN, whereas only 4.10% of HT29 DEGs were hits in the ISN (Figure 3C,D; Table 1).

#### 2.4.2. Docking and Transcript-Based Hub Analysis and Network Spanning Results

The integrated protein list of dockable proteins (DPs) and DEG protein lists predicted the number of hits for each organ network hub (Figure 4A,B). Of the proteins, 26.19% hit hubs in the BSN (Appendix A, analysis column) and 20.27% hit hubs in the ISN (Appendix A, analysis column; Table 2). The permutation test results showed that the major protein hits in the BSN hub were significant (*p* = 0.04705). Furthermore, the ISN results were statistically significant, unlike the results for the NP proteins and DPs (*p* = 0.02332) (Table 2).

Network spanning describes the activity of organs by defining each node in the network as a dimension and determining how populated the network dimension is with drug-interacting proteins. Although statistical significance was confirmed for the analysis using DPs and DEGs in both networks, network spanning analysis through first-shell interactions showed that the BSN spanned 67.12%, whereas the ISN spanned only 51.09% (Figure 4C,D). Furthermore, the BSN was able to achieve full network spanning in only three walks, whereas the ISN required six additional walks.

#### 2.4.3. Over-Representation Analysis (ORA) and Gene Set Enrichment Analysis (GSEA) Results Using Transcriptome Analysis Results

Protein information for analysis was extracted from the BSN, which was predicted to be more dominantly affected by ACD. ORA was performed by combining DPs and DEGs, and 230 proteins were used. ORA using ACD showed that the calcium signaling pathway had the highest score in the KEGG pathway, and excitatory chemical synaptic transmission had the highest score in the Gene Ontology (GO) biological process (Figure 5A,B). In addition, ACD was predicted to be involved in psychotropic drug addiction and neurotransmitter activation. The results of the ORA for 698 proteins across the BSN dimensions were generally similar but slightly different (Appendix A). It seemed to focus more on psychotropic drugs and neurotransmitters than on calcium signaling.

The GSEA results using transcriptome expression in SW1783 (Appendix A) showed an overall pattern of reduced inflammation, including decreased MAPK and TGF-β signaling pathways, and increased phagocytosis. Notably, ACD has been predicted to activate long-term potentiation (LTP). Although the statistical significance was low at high doses, the calcium signaling pathways predicted to be activated in ORA were also activated. The GSEA results using the transcriptome expression of HT29 cells (Appendix A) showed that the terms related to the digestive system, such as diabetes, differed from the SW1783 analysis results. However, LTP was also activated in the same manner as in SW1783.

## 3. Discussion

We constructed the BSN and ISN based on bioinformatics and predicted that ACD would have a more dominant effect on the brain than on the intestine using NP and large-scale molecular docking analysis. Through validation experiments using transcriptomes, we confirmed that the transcriptome expression in brain-based cell lines was significantly higher than that in intestine-based cell lines. In addition, through a novel transcriptome-based analysis, we confirmed that ACD spanned the dimensional space of the BSN dimension more easily than that of the ISN dimension. This study is expected to be highly applicable for predicting the sites of action of complex compounds, including natural products.

We constructed an organ network using CoSE and predicted the action points of the organ network using multifaceted analysis (Figure 1 and Figure 2). The spring-embedder layout is an algorithm that simulates a system connected by springs, assuming that the nodes have electric charges [17]. In this study, we cross-checked not only the intuitive calculation method of degree centrality but also the hub of the nested intergraph constructed based on electric repulsion. Although the proteins obtained through the docking analysis were not determined to have hub properties with a simple degree centrality, we confirmed that they could have hub properties using CoSE (Figure 2A). In this study, we used a research method that approaches the properties of networks in terms of spring connection centrality and degree centrality and expect that this multi-faceted analysis will contribute to the development of natural product-based NP analyses.

In this study, the prediction results were verified through transcriptome validation experiments, and the in silico prediction was enhanced using docking analysis (Figure 3 and Figure 4). Docking analysis can predict direct interactions between compounds and targets, and the transcriptome can identify on- and off-target effects through transcriptome changes after protein signaling [18]. Although both analyses are good bioinformatics-based methods, there are limitations to using only one method because the time points used for analysis are different. To overcome these limitations, two complementary analytical methods must be used [19]. In this analysis, there were few overlapping proteins between the DPs and DEGs. Nevertheless, although the individual results of the docking and transcriptome analyses of the ISN were not statistically significant, the results of the combined data were statistically significant. This result serves as a basis for demonstrating that the two datasets can be used complementarily, and this approach is expected to improve current bioinformatics analyses.

In addition, by presenting new results using transcriptome validation experiments, we were able to add evidence that ACD is predominantly applied to the brain rather than to the intestine (Figure 3 and Figure 4). Docking analysis is the starting point of protein signaling, and transcriptome analysis is the final goal of protein signaling; therefore, both are closely related to protein signaling. Regarding protein signaling, including neighboring nodes of docking and transcriptome analyses may result in an analysis target that is more precise [20,21]. Moreover, if the proteins that make up the network perform even slightly independent roles, the network can be said to have a dimensional space equal to the number of nodes that make up the network [22]. Based on this, we can determine how much the network dimension has spanned and how quickly it can be fully spanned through “walks” on the network [23,24]. Although this analysis method requires further verification, a new idea based on the logic of linear algebra can become a new method for NP analysis in the future.

In this study, we derived the results of transcriptome-based ORA and GSEA, which showed that ACD affects calcium signaling and LTP. Additionally, the docking analysis confirmed that the components of ACD act on neurotransmitters and calcium channels (Appendix A). LTP is a crucial mechanism of synaptic plasticity and is essential for learning and memory formation [25]. Specifically, calcium signaling activates N-methyl-D-aspartate (NMDA) receptors, which increase the calcium concentration in postsynaptic neurons, thereby inducing LTP [26]. Various studies have shown that ACD regulates its activity through calcium-signaling pathways [27,28,29]. Additionally, ACD inhibited AD through a complex regulatory network centered on GRIN1 and MAPK1 [30]. GRIN1 is a key component of NMDA and plays a critical role in synaptic plasticity and memory formation, whereas mitogen-activated protein kinase 1 (MAPK1) is involved in neuronal survival and synaptic plasticity. These findings suggest that ACD positively affects long-term memory formation through its neuroprotective effects. Moreover, it has been revealed that the ACD polysaccharide exhibits antidepressant effects in rats [30]. These antidepressant effects manifest through improvements in neural plasticity and synaptic function, which can also positively affect long-term memory formation. Therefore, it is highly likely that the ACD polysaccharides contribute to neuroprotection and memory formation. These findings suggest that ACD influences LTP by modulating calcium signaling, indicating its potential as a promising candidate for the treatment of Alzheimer’s disease and other neurodegenerative disorders. In particular, CHRM1, RPE65, CHRM4, GRIK4, CACNA1B, and CACNA1E, predicted to have high probabilities of interaction in the docking analysis, are expected to be prioritized in future studies.

As mentioned in the Introduction, ACD is an herbal medicine that can be used to treat digestive problems, including diarrhea [10]. However, if toxic alkaloid compounds contained in ACD, such as aconitine, are consumed in large quantities, they can cause diarrhea by strong contraction of the ileum, which can interfere with treatment [10]. Although ACD may cause diarrhea, our analysis suggests that it has therapeutic potential because it is regulated by actions that occur in the brain and not in the intestine. The pharmacological mechanism of ACD, which has traditionally been used, can be reinterpreted as improving diarrhea by improving neurotransmitter dysfunction in the brain [11]. The research method proposed to identify the dominant organ will not only reinterpret traditional medicine that has not yet been fully elucidated but can also explain the ‘meridian affinity’ theory that will help advance traditional medicine.

However, this study had some limitations. First, the proposed organ network is limited to physiological networks and cannot reproduce all pathological situations. Our presented model is at an early stage and is not yet suitable for application in clinical practice. Therefore, generating a model that can consider the blood–brain barrier and ADMET (Absorption, Distribution, Metabolism, Excretion, Toxicity) is necessary in further studies. Second, the number of organs that can be studied is limited owing to the difficulty in conducting large-scale transcriptomic verification. Since ACD is highly toxic, side effects may occur. The liver and kidneys are the most important organs for evaluating drug toxicity. Because this study focused on the pharmacological action, the scope was limited to the gut–brain axis. Therefore, further studies are necessary to expand the model to the liver and kidneys and assess possible side effects of the drug. Nevertheless, the method proposed in this study can identify the organs in which a specific drug preferentially acts. In particular, the consistency of the analysis results was confirmed using various bioinformatics techniques, such as NP, docking, transcriptome, and docking-transcriptome complex analyses, thereby increasing the predictive power of the analysis results. This method will be important in the future to identify which organs show the dominant effect, not only among natural products but also among drugs, and can also be utilized for exploring drug side effects. In particular, the method for predicting the pharmacological mechanism of action on organs using an in silico model may be clinically significant, as it can considerably save time and cost and play an important role in ensuring drug safety.

## 4. Materials and Methods

### 4.1. Organ Network Construction

To identify the organs in which ACD exerts its predominant pharmacological effect, the BSN and ISN were constructed. The BSN and ISN were constructed using the proteins predominantly expressed in each organ. Information on the proteins predominantly expressed in each organ was obtained from the HPA database (https://www.proteinatlas.org/humanproteome/tissue, accessed on 16 May 2024) [31]. The BSN was constructed using all brain-specific tissues provided by the HPA database, and the ISN was constructed using intestine-specific tissues.

A protein-based organ network was constructed using the STRING database (v12; https://string-db.org/, accessed on 5 March 2024) [32]. Network nodes were defined as proteins, and edges were defined as interaction scores between two proteins greater than or equal to 700 among the interaction scores provided by the STRING database. After constructing the network, a fraction with fewer than 10 nodes was excluded from the organ network to define the organ network. The constructed network was centered on high-order hub proteins using the CoSE layout [17]. Data preprocessing was performed using Python software (v3.10.12), and network construction and visualization were performed using Cytoscape software (v3.10.1) [33].

### 4.2. NP Analysis

The NP analysis was performed using the BATMAN-TCM platform (v1, http://bionet.ncpsb.org.cn/batman-tcm/index.php/Home/Index/index, accessed on 22 May 2024) [34], which is a well-known and frequently used analytical platform. ACD was searched on the BATMAN-TCM platform, and all compound–target interaction prediction results derived from the search were downloaded. Among the downloaded predicted interactions, those with a score cutoff of 10 or higher were selected as valid interactions of the ACD compounds, and the proteins included in the valid interactions were selected as NP-based interacting proteins.

We projected the NP-based interacting proteins onto the BSN and ISN to predict the extent to which the compounds of ACD interacted with the hub proteins of each organ network. We computed the hit ratio between the hub proteins with the top 20% degree of centrality in each network and obtained the statistical significance for the abundance acting on the hubs using 100,000 permutation tests [35]. The hit ratio between the hub proteins with the top 20% degree centrality in each network and ACD was calculated. If one compound interacted with a protein that constituted the organ network hub, the compound and protein were defined as hits. The statistical significance of the interactions of ACD compounds with hub proteins was calculated using 100,000 permutations. The permutation test randomly selects the same number of nodes as the projected NP-based interacting proteins in the network and checks the ratio of hub proteins among the selected nodes. After 100,000 repetitions, the rank of the actual NP hub protein ratio among the virtual hub protein ratios was assessed to calculate the *p*-value. Based on the above results, we compared the hub protein hit ratios of the ACD components in the BSN and ISN to confirm the prediction results regarding in which network the drug may have a more dominant pharmacological action. Data preprocessing was performed using Python software (v3.10.12), and the network centrality calculations and visualization of NP-based interacting protein projection results were performed using Cytoscape software (v3.10.1).

### 4.3. Large-Scale Molecular Docking Analysis

To identify the organ system in which the pharmacological effect of ACD is dominant, we performed large-scale molecular docking analysis [19]. The compound information included in ACD was collected using the following three natural product compound databases: TCMSP [36], BATMAN-TCM, and TM-MC [37]. To use only compounds whose 3D structures were confirmed among the collected compound, the structures of the compounds were downloaded from the PubChempy library [38]. Compounds with confirmed 3D structures were designated as ACD compounds for docking. The proteins used in the docking analysis were selected based on the following criteria: first, we targeted 854 druggable proteins provided in the HPA database (https://www.proteinatlas.org/humanproteome/tissue/druggable, accessed on 3 May 2024); and second, we restricted our analysis to proteins included in the BSN and ISN. We selected a list of druggable proteins that met both criteria and downloaded the 3D structures of the selected druggable proteins from the AlphaFold database (v2) [39].

Preprocessing for the docking analysis was performed using Python software (v3.10.12) and the OpenBabel Python library (v3.1.0) [40]. Both the 3D structures of the ACD compounds and the druggable proteins were converted to pdbqt form for docking analysis. Docking analysis was performed using the AutoDockVina Python library [41] for all ACD compounds and druggable protein pairs in each organ. The parameters used in the docking analysis were as follows: center = (0, 0, 0), box size = (126, 126, 126), and exhaustiveness = 100. Binding affinity scores of <−8 kcal/mol were selected as valid interactions and used in the docking analysis results. Among the ACD compounds, proteins with >10 interactions were selected as DPs with high interaction potential. The hit ratio and permutation test of the hub proteins with the top 20% degree centrality were performed in the same manner as the NP analysis. The docking analysis results were visualized using Cytoscape software (v3.10.1).

### 4.4. Producing ACD-Induced Transcriptomes in SW1783 and HT29 Cell Lines

#### 4.4.1. Chemicals and Reagents

Dulbecco’s modified Eagle medium (DMEM), phosphate-buffered saline (PBS), TrypLE Express, penicillin–streptomycin, and fetal bovine serum (FBS) were purchased from Gibco (Grand Island, NY, USA). Leibovitz’s L-15 medium was purchased from the American Type Culture Collection (ATCC, Manassas, VA, USA). The cell culture flasks and multiwell culture plates were purchased from Thermo Fisher Scientific (Waltham, MA, USA). Dimethyl sulfoxide (DMSO) and wortmannin were purchased from Sigma-Aldrich (St. Louis, MO, USA), and QIAzol Lysis Reagent was purchased from Qiagen (Hilden, Germany). The EZ-Cytox Cell Viability Assay Kit was purchased from DoGen Bio (Seoul, Republic of Korea).

#### 4.4.2. Preparation of Hot Water Extracts of ACD

Dried ACD was supplied by Kwangmyungdang Medicinal Herbs Co. (Ulsan, Republic of Korea), and its morphology was carefully validated by Dr. Goya Choi from the Herbal Medicine Resources Research Center, Korea Institute of Oriental Medicine (KIOM), Republic of Korea. A voucher specimen (2-23-0011) was deposited in the Korean Herbarium of Standard Herbal Resources of KIOM (Naju, Republic of Korea). The crushed material was extracted at 100 ± 3 °C for 3 h using a water reflux system (MS-DM609, Misung Scientific, Yangu, Republic of Korea) and filtered through a 5 µm cartridge filter (KOC Biotech, Daejeon, Republic of Korea). The filtrates were concentrated at 40 °C using a rotary evaporator (Ev-1020t, SciLab, Seoul, Republic of Korea) and freeze-dried (LP20, ilShin Biobase, Dongducheon, Republic of Korea). The yield of water extract of ACD was 25.4%, and the final extracts were homogenized and stored in an airtight container in a cold room at 4 °C. For the in vitro investigation, homogenous ACD extract was vigorously vortexed for 30 min at room temperature in PBS (Thermo Fisher Scientific) containing 2% DMSO and then sterilized by filtering through a 0.22 µm membrane. The stock solution of ACD extracts (10 mg/mL) was aliquoted in a 1.5 mL tube and stored at −80 °C until use.

#### 4.4.3. Cell Culture

The human astrocytoma cell line SW1783 (HTB-13) and human colorectal adenocarcinoma cell line HT29 (HTB-38) were purchased from ATCC (Manassas, VA, USA). SW1783 cells were maintained in Leibovitz’s L-15 medium supplemented with 10% (*v*/*v*) heat-inactivated FBS, 100 IU/mL penicillin, and 100 mg/mL streptomycin at 37 °C without CO_2_. HT29 cells were maintained in DMEM supplemented with 10% (*v*/*v*) heat-inactivated FBS, 100 IU/mL penicillin, and 100 mg/mL streptomycin at 37 °C in a 5% CO_2_ incubator. The cells were subcultured every 3–4 days depending on the cell density.

#### 4.4.4. Drug Treatment

One day before drug treatment, SW1783 and HT29 cells were plated at 1.5 × 10^5^ and 5 × 10^5^ cells/well, respectively, in a six-well plate containing 3 mL of growth medium. The cells were exposed to 20 (Low), 100 (Intermediate), and 500 (High) µg/mL of ACD by treatment with 150 µL per well of 0.4, 2, and 10 mg/mL ACD extracts. To assess the appropriate drug concentrations for treatment, we performed cytotoxicity tests to determine the drug doses that maintained 80% cell viability (IC20). These doses were then adopted as the maximal doses for treatment and RNA sequencing (RNA-seq) analysis. For drugs with an undetermined IC20, the highest treatment concentration was set to 500 µg/mL for extracts, considering both their solubility and clinical relevance. To confirm the influence of concentration, cells were treated with drugs at three different concentrations using 1/5 serial dilutions, thereby exposing them to low, medium, and high drug doses. No cytotoxicity was observed at a high dose (500 µg/mL), which was confirmed using an EZ-Cytox cell viability assay kit. PBS with 2% DMSO was used as the vehicle, and wortmannin was treated at a concentration of 10 µM as a positive control. After 24 h of drug treatment, the cells were washed three times with ice-cold PBS. Total cell lysates were prepared using QIAzol Lysis Reagent and stored at −70 °C until RNA extraction.

#### 4.4.5. RNA Preparation for RNA-Seq

SW1783 and HT29 cells were subjected to total RNA extraction using QIAzol Lysis Reagent according to the manufacturer’s instructions. The concentration of the isolated RNA was determined using an Agilent RNA 6000 Nano Kit (Agilent Technologies, Waldbronn, Germany). The RNA concentration was determined using a Quant-it RiboGreen RNA Assay Kit (R11490, Thermo Fisher Scientific), and the RNA quality was assessed by determining the RNA integrity number (>7) and 28S:18S ribosomal RNA ratio (>1.0) using a 2100 Bioanalyzer Instrument (Agilent Technologies, Waldbronn, Germany). Total RNA was processed to prepare an mRNA sequencing library using the MGIEasy RNA Directional Library Prep Set (#1000006386; MGI Tech, Shenzhen, China), according to the manufacturer’s instructions. The library was quantified using the QuantiFluor ssDNA System (E3190; Promega, Madison, WI, USA). The prepared DNA nanoballs were sequenced on an MGISeq system (MGI Tech, Shenzhen, China) with 100 bp paired-end reads.

#### 4.4.6. RNA-Seq Preprocessing Method

The quality of the raw RNA-seq data was assessed using FastQC (v0.11.9) [42]. Adapter sequences were removed from the reads using Trim Galore (v0.6.6) [43]. The cleaned reads were aligned to the human reference genome GRCh38 (hg19) using STAR (v2.7.9.a) [44]. Transcript abundance per gene was quantified using RSEM (v1.3.3) [45] with the gene annotation GRCh38.84. The expected read counts and transcripts per million were used as the gene expression levels for further analyses. Differential gene expression between the treatment and vehicle groups was analyzed using the Wald test implemented in the R package DESeq2 (v1.38.2) [46]. The RNA-seq data were analyzed using R (v4.3.1). RNA sequence data were deposited in the Gene Expression Omnibus under the accession number GSE273185 (https://www.ncbi.nlm.nih.gov/geo/query/acc.cgi?acc=GSE273185, accessed on 7 August 2024).

### 4.5. Transcriptome Verification Experiments and Analysis

#### 4.5.1. DEG Selection and Comparison of the Number of DEGs

Through transcriptome experiments, the DEGs showing significant changes in expression levels in each cell line were selected and compared. DEGs greater than log2(1.5) or less than −log2(1.5) were selected as DEGs with an adjusted *p*-value < 0.05. The number of DEGs was confirmed at three concentrations of wortmannin, which was used as a positive control, and in two cell lines, SW1783 and HT29. In addition, DEGs obtained by the high-concentration treatment of SW1783 and HT29 cells were projected onto each organ network to identify the action point in the network. DEG information extraction was performed using R software (v4.3.1), and the visualization of the action points was performed using Cytoscape software (v3.10.1).

#### 4.5.2. Docking and Transcript-Based Hub Analysis and Network Spanning Analysis

We performed additional analyses to determine whether the combined DPs and DEGs derived from the network performed better. We identified action points by combining DPs and DEGs derived from each organ network and checked how many times the action points hit the hub proteins in each organ network. As in the previous analysis, we performed a permutation test using the hit ratio of hub proteins with the top 20% degree centrality to determine statistical significance.

The concept of full-spanning a dimension is a concept used in linear algebra. In order to express three dimensions, we need axes that play independent roles, namely, the x-axis, y-axis, and z-axis. Using this concept, in this study, we considered the BSN and ISN to have 1040 and 366 dimensions, respectively, and attempted to see how quickly they can span the dimensions to express the 1040 and 366 dimensions (Appendix A). In other words, the proteins that make up the BSN and ISN are proteins that play independent roles, and the assumption is that signals must be transmitted to all nodes in the ‘functional space’ of the organ in order for the organ to perform its full function.

After predicting how DPs and DEGs interacted with hub proteins, we investigated the extent to which proteins could span the dimensions of the organ network. First, we extracted the list of first-shell interaction nodes, which are neighbors of the DPs and DEGs in each network. We mapped the first-shell interaction nodes to the network along with the DPs and DEGs to determine the extent to which they span the network dimension in the entire network. Finally, we performed an analysis to determine the number of steps needed to fully span the network, that is, how far apart the nodes with the farthest distances from the DPs or DEGs were. Analysis was performed using Python software (v3.10.12) and Networkx library (v3.2.1) [47], and visualization was performed using Cytoscape (v3.10.1).

#### 4.5.3. ORA and GSEA Results Using Transcriptome Analysis Results

The efficacy of ACD in the dominant organs was predicted using the transcriptome analysis results. First, ORA was performed using the protein list that (1) integrated the DPs and DEGs that hit the BSN and (2) the network-spanning protein list that considered neighboring nodes in the EnrichR platform [48]. ORA was performed using the gene lists of KEGG pathways [49] and GO biological processes [50]. The top 10 ORA results with the highest combined scores provided by the EnrichR platform were selected. GSEA was performed using the transcriptome expression results from SW1783 and HT29 cells treated with high doses. GSEA was performed for curated KEGG pathway gene sets in the Molecular Signature Database (MSigDB v7.5.1) [51] using the fgsea package (v1.24) [52] in R with parameters of minimum size 15, maximum size 500, and 100,000 permutations. The statistical significance of the GSEA results was evaluated by adjusting the *p*-value using the Benjamini–Hochberg procedure (*p* < 0.0001) [53]. The GSEA results were visualized as a heatmap using Pheatmap (v.1.0.12). Only pathways with results satisfying at least one statistical significance for the three concentrations and wortmannin were included in the heatmap. The KEGG pathway gene list was used for GSEA.

### 4.6. Phytochemical Analysis of ACD Extract by LC/MSMS

The main components of the ACD extract used in the experiment were analyzed using LC/MSMS. In brief, the 3 major ingredients of the ACD extract were analyzed with a Xevo TQD (Waters, Milford, MA, USA) equipped with an Acquity UPLC BEH C18 (1.7 μm, 2.1 × 100 mm) column. The standard chemicals used in the analysis were purchased from Chemfaces (Wuhan, Hubei, China). The gradient conditions for chromatographic separation using 0.1% formic acid in water (A) and acetonitrile (B) were as follows: 0–0.1 min, 20% B; 0.1–14 min, 20–95% B; 14–15 min, 95–100% B; 15–15.1 min, 100–20% B; 15.1–18 min, 20% B; and equilibration with 20% B for 4 min at a flow rate of 0.3 mL/min. The temperature of the column was 45 °C, and the auto-sampler was maintained at 5 °C. The injection volume of each sample was 2 μL. The MS/MS data for qualitative analysis were processed to identify and confirm components in the extracts based on the retention time, accurate m/z value, isotope distribution, and fragment ions compared to reference standards. The reference standards for each target compound were analyzed, and the amounts of compounds were quantified using corresponding calibration curves of reference standard compounds. The results of each analysis are summarized in Appendix A.

## Figures and Tables

**Figure 1 ijms-25-10219-f001:**
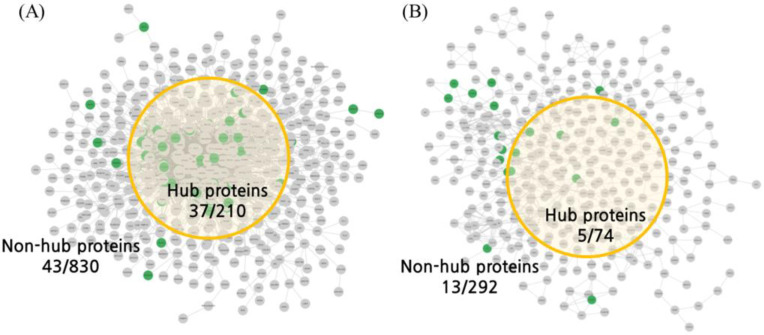
Action points of the organ networks derived from NP analysis. Proteins interacting with ACD were predicted by NP analysis, and the predicted proteins were projected to the BSN and ISN. The nodes of the network are the proteins that constitute the BSN and ISN, and the edges of the network are the connections with interaction scores of 700 or more in the STRING database. Green nodes represent interacting proteins in the network pharmacology analysis results, and yellow circles represent the locations of proteins that act as hubs in the network. (**A**) BSN and interacting proteins projected to BSN. (**B**) ISN and interacting proteins projected to ISN. NP, network pharmacology; ACD, *Aconitum carmichaeli* Debeaux; BSN, brain system network; ISN, intestinal system network.

**Figure 2 ijms-25-10219-f002:**
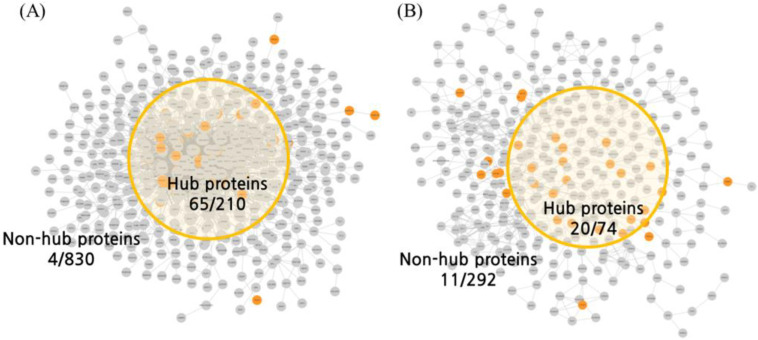
Action points of the organ networks derived from MD analysis. Using large-scale MD analysis, proteins interacting with ACD were predicted, and the predicted proteins were projected to the BSN and ISN. The nodes of the network are the proteins that constitute the BSN and ISN, and the edges of the network are the connections with interaction scores of 700 or more in the STRING database. Orange nodes are dockable proteins that have more than 10 hits in the docking analysis results, and yellow circles indicate the positions of proteins that act as hubs in the network. (**A**) BSN and dockable proteins projected to BSN. (**B**) ISN and dockable proteins projected to ISN. MD, molecular docking; ACD, *Aconitum carmichaeli* Debeaux; BSN, brain system network; ISN, intestinal system network.

**Figure 3 ijms-25-10219-f003:**
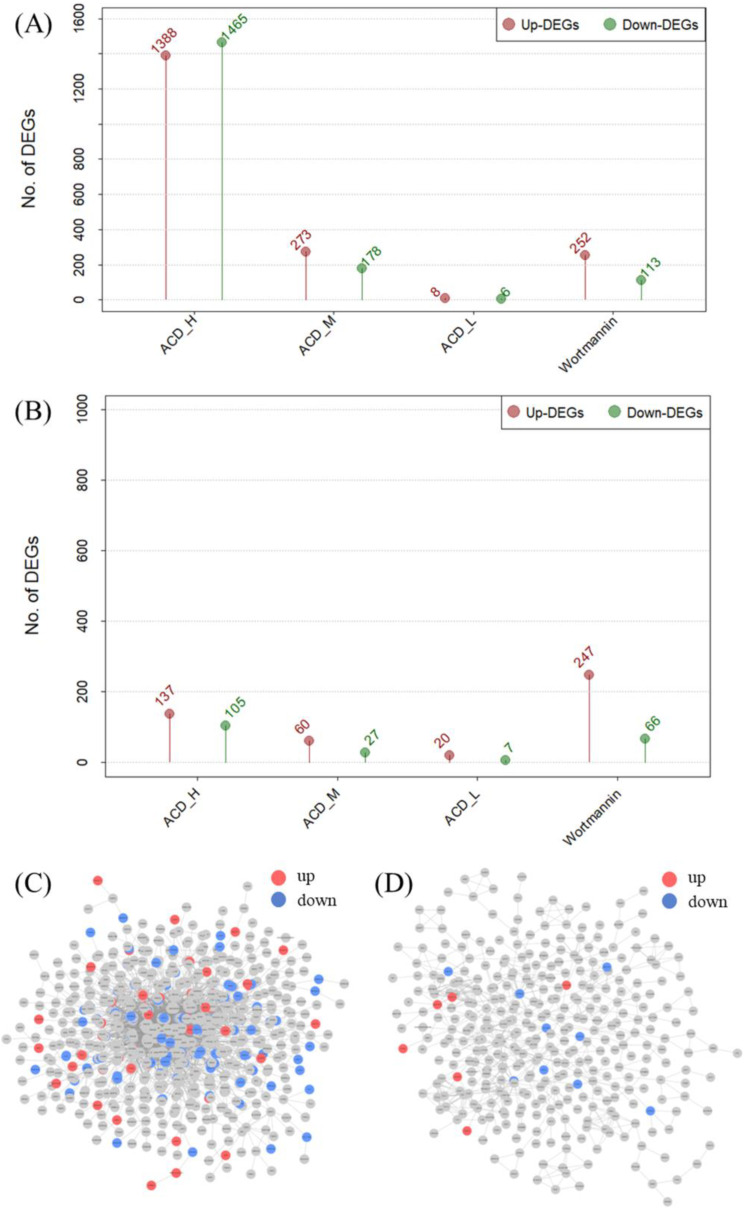
Results of validation experiments using transcriptomes. The number of DEGs whose expression levels were changed after treatment with ACD in the SW1783 and HT29 cell lines was identified and visualized. The water extract of ACD was administered at high doses (500 µg/mL), medium doses (100 µg/mL), and low doses (20 µg/mL), and wortmannin was administered as a positive control. The results of the number of DEGs were compared. (**A**) The number of DEGs expressed after ACD was administered to the SW1783 cell line. (**B**) The number of DEGs expressed after ACD was administered to the HT29 cell line. In (**A**,**B**), the upregulated DEGs are indicated by red lines, and the downregulated DEGs are indicated by green lines. (**C**) DEGs of SW1783 projected to the BSN. (**D**) DEGs of HT29 projected to the ISN. In (**C**,**D**), upregulated DEGs are represented by red nodes, and downregulated DEGs are represented by green nodes. DEG, differentially expressed gene; ACD, *Aconitum carmichaeli* Debeaux; BSN, brain system network; ISN, intestinal system network.

**Figure 4 ijms-25-10219-f004:**
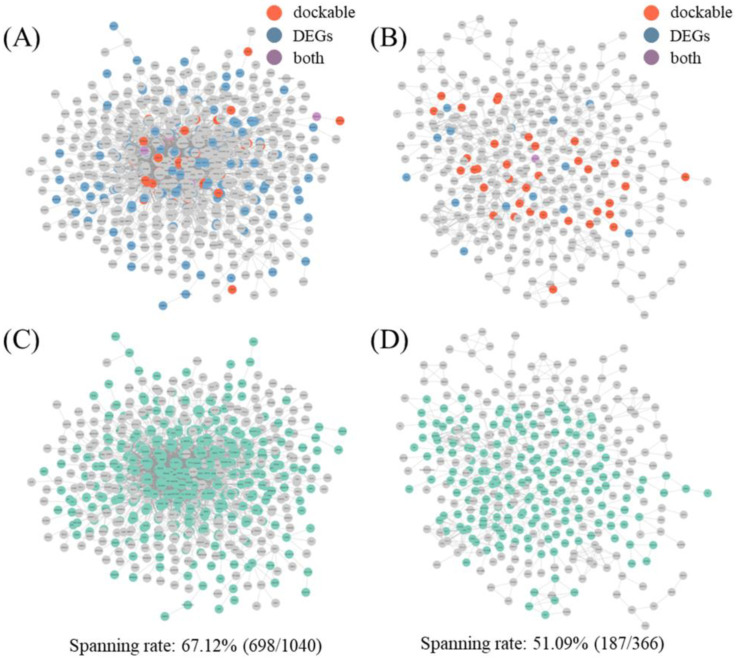
Results of network spanning analysis using transcriptome data. The results of the analysis performed using transcriptome data derived after administration of ACD to the SW1783 and HT29 cell lines are visualized. (**A**) Visualization of brain DP and DEG projections to BSN in SW1783 cell line. (**B**) Visualization of intestine DP and DEG projections to ISN in HT29 cell line. In (**A**,**B**), red nodes are DPs, blue nodes are DEGs, and purple nodes correspond to both. (**C**) Analysis of spanning rate using DPs, DEGs, and neighbor nodes of the integrated protein list in BSN. (**D**) Analysis of spanning rate using DPs, DEGs, and neighbor nodes of the integrated protein list in ISN. Green nodes in (**C**,**D**) include all action points, DPs, DEGs, and neighbor nodes in the network. The spanning rate described below the figure represents the ratio of action points to the total number of nodes in the network. ACD, *Aconitum carmichaeli* Debeaux; DP, dockable protein; DEG, differentially expressed gene; BSN, brain system network; ISN, intestinal system network.

**Figure 5 ijms-25-10219-f005:**
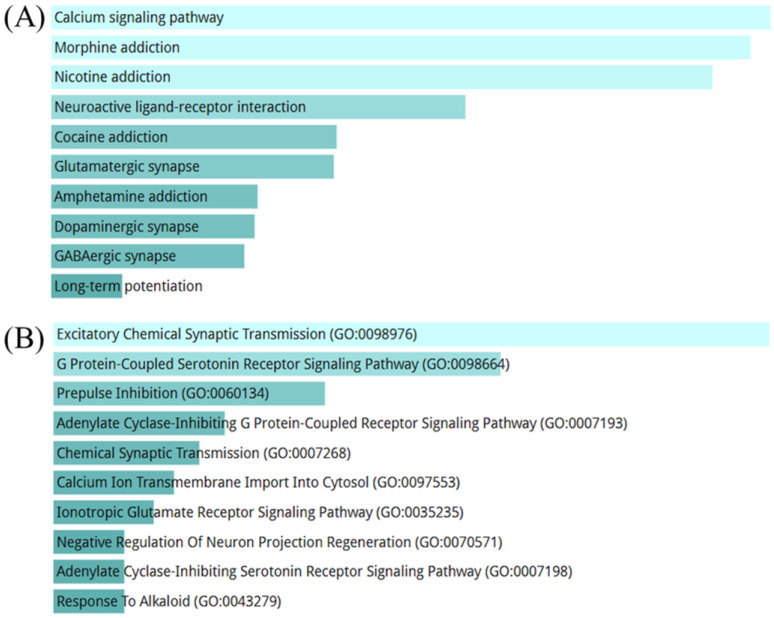
ORA results using action points of the BSN. ORA results performed on the BSN using DPs of ACD that act on the brain proteins and ACD-induced DEGs of the SW1783 cell line. The longer the bar graph and the lighter the color, the higher the statistical significance of the term. (**A**) ORA results using the KEGG pathway gene set. (**B**) ORA results using the GOBP gene set. ORA, over-representation analysis; BSN, brain system network; DP, druggable protein; ACD, *Aconitum carmichaeli* Debeaux; DEG, differentially expressed gene; KEGG, Kyoto Encyclopedia of Genes and Genomes; GOBP, Gene Ontology biological process.

**Table 1 ijms-25-10219-t001:** Protein hit results in organ networks.

Analysis	Organ	#Whole Proteins	#Hit Proteins	Hit Ratio
Network pharmacology	BSN	1040	80	7.69%
	ISN	366	18	4.92%
Molecular docking	BSN	1040	69	6.39%
	ISN	366	31	8.47%
DEG count	BSN	1040	172	16.54%
	ISN	366	15	4.10%

DEG, differentially expressed gene; BSN, brain system network; ISN, intestinal system network. # is a number sign and indicates the number of proteins in the table.

**Table 2 ijms-25-10219-t002:** Hub protein hit results in organ networks.

Analysis	Organ	CCM	#Hub	HDT	#Hit	Hit Ratio	*p*-Value
NP	BSN	Degree	210	>11	37	17.62%	<0.00001
	ISN	Degree	74	>6	5	4.92%	0.13133
MD	BSN	Degree	210	>11	28	13.33%	0.00004
		CoSE		-	65	30.95%	-
	ISN	Degree	74	>6	5	13.51%	0.08452
		CoSE		-	20	27.03%	-
IP	BSN	Degree	210	>11	55	26.19%	0.04705
	ISN	Degree	74	>6	15	20.27%	0.02332

CCM, centrality calculation method; #Hub, whole hub count; HDT, hit degree threshold; #Hit, hit hub count; NP, network pharmacology; MD, molecular docking; IP, integrated protein list of dockable proteins and differentially expressed genes; BSN, brain system network; ISN, intestinal system network; CoSE, Compound Spring Embedding. Cells without values are treated with a hyphen symbol. # is a number sign and indicates the number of proteins in the table.

## Data Availability

The raw sequences and processed data were deposited in the NCBI Gene Expression Omnibus (GEO, https://www.ncbi.nlm.nih.gov/geo/query/acc.cgi?acc=GSE273185) with accession number GSE273185.

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
