# Peer review of "Discovery and Prediction Study of the Dominant Pharmacological Action Organ of Aconitum carmichaeli Debeaux Using Multiple Bioinformatic Analyses"

_ijms, 2024, doi:10.3390/ijms251810219_

Round 1
Reviewer 1 Report (New Reviewer)
Comments and Suggestions for Authors
dear author
Overall, the work on "Discovery and prediction study of the dominant pharmacological action organ of Aconitum carmichaeli Debeaux using multiple bioinformatic analyse". the researcher constructed Brain system network and intestine network and identified the ACD results uisng NP analysys and he concludes SW1783 cell line had more differentially expressed genes than HT29 cell line, it gives ACD preferentially acts in the brain, very nice work, i have my small suggestions are:
Include images of Aconitum carmichaeli Debeaux and molecular structures.
you mentioned about ‘Meridian affinity' but no flow chart.
BATMAN-TCM results are good but hub protein not understanding and spanning rate. Please provide a more detailed explanation.
dedicated page for the network spanning analysis data image would greatly give clear under stand.
provide Experimental validation images
you mentioned about 3D structures i didnt find
Please ensure that Table S9 (LC/MSMS) is readily available.
Author Response
Reviewer 1.
Overall, the work on "Discovery and prediction study of the dominant pharmacological action organ of Aconitum carmichaeli Debeaux using multiple bioinformatic analyse". the researcher constructed Brain system network and intestine network and identified the ACD results uisng NP analysys and he concludes SW1783 cell line had more differentially expressed genes than HT29 cell line, it gives ACD preferentially acts in the brain, very nice work, i have my small suggestions are:
Response: Thank you for taking the time and effort to review our manuscript, and for your encouraging comments regarding our study.
- Include images of Aconitum carmichaeli Debeaux (ACD) and molecular structures.
Response: In this study, we focused on presenting new research methods, and natural products were used as examples that can be applied to new research methods. Based on the reviewer's comments, ACD and the structures of main aconitine-type diterpene alkaloids were presented in a figure, but were judged to be tangential to the primary aims of this study, so they were instead included as Supplementary Figure 1.
Figure S1
- you mentioned about ‘Meridian affinity' but no flow chart.
Response: “Meridian affinity” is a “therapeutic theory” used in traditional Asian medicine (TAM). It differs from general medical knowledge, and posits that organs where crosstalk can occur more easily can be grouped into a system to show efficacy together. Among them, ACD, which was used as one of the main adaptogens, is known to act on the spleen meridian and heart meridian among the 12 organ-crosstalk systems. This theory is called meridian theory because it is believed that organ-crosstalk occurs due to the flow in the human body known as "Qi", and in the past, this was tried to be proven through observation of patients. The introduction section discusses meridian theory as natural adaptogens in the past suggested that an organ that can act with this theory, and currently, the purpose is to identify a new organ of action using biological data. A flow chart on meridian affinity can be presented for the purpose of academic curiosity in the review, but it does not fit the theme of this study, which is a comparison of the gut and the brain. We have attached a figure that briefly summarizes meridian affinity.
- BATMAN-TCM results are good but hub protein not understanding and spanning rate. Please provide a more detailed explanation. dedicated page for the network spanning analysis data image would greatly give clear understand.
Response: In network theory, the concept of walk is an important concept for analysis. The concept of spanning refers to the dimensions that can be expressed in linear algebra, and in simple terms, it refers to how diverse the expressions can be. We modeled the virtual brain and gut as a network using proteins that are expressed more dominantly than other organs. In this situation, ‘spanning’ refers to the functions of the brain and gut that can be fully exerted. In other words, if all the important proteins in the brain and gut are touched, it is assumed that the ACD is likely to act on the corresponding organ. Therefore, we calculated how efficient it is to touch all the nodes with independent functions in the network. At the request of the reviewer, we added an additional explanation to the text along with a schematic diagram of network spanning (Supplementary Figure 7).
Revised manuscript: The concept of full-spanning a dimension is a concept used in linear algebra. In order to express three dimensions, we need axes that play independent roles, namely the x-axis, y-axis, and z-axis. Using this concept, in this study, we considered the BSN and ISN to have 1,040 and 366 dimensions, respectively, and attempted to see how quickly they can span the dimensions to express the 1,040 and 366 dimensions (Figure S7). In other words, the proteins that make up the BSN and ISN are proteins that play independent roles, and the assumption is that signals must be transmitted to all nodes in the ‘functional space’ of the organ in order for the organ to perform its full function. (Lines 472–479)
Figure S7:
- provide Experimental validation images
Response: We performed experimental validation by producing transcriptome information. If it were an in vivo/vitro experiment for actual efficacy verification, it would be possible to submit quantitative images of cell progress or kinase levels. However, it is very difficult to present this experiment in this manner. We could also present new analysis results, such as volcano plots, on the experimental results, but since we are preparing additional studies on them, it is difficult to disclose this data in advance in this paper. Instead, we will send you the cell pictures that generated the transcriptomes that we performed for verification through review.
HT29 – DMSO HT29 – ACD high dose
SW1783 – DMSO SW1783 – ACD high dose
- you mentioned about 3D structures i didnt find
Response: The 3D structure we used in this study refers to "the 3D structure of the protein" and "the 3D structure of the compound" used in the docking analysis. The protein 3D structure can be found in the AlphaFold 2.0 database (https://alphafold.ebi.ac.uk/) that we used in our study, and the compound structure can be found in the PubChem database (https://pubchem.ncbi.nlm.nih.gov/). We do not think that presenting the 3D structure is in line with the research topic of this paper. It is a tool used to predict the interactions that can occur in each organ, and this was confirmed at the system level. Therefore, presenting each compound and protein structure is not an efficient strategy. Instead, we have attached an example of the analysis result of the 3D structure:
Aconitine – CACNB4 interaction results (-7.006 kcal/mol)
- Please ensure that Table S9 (LC/MSMS) is readily available.
Response: We are not sure what the reviewer intended. We think it is probably the peak diagrams for the LC/MSMS. If this was not what you intended, please give your opinion on this matter. We have added the peaks of LC/MSMS to Table S9.
Supplementary Table 9. Quantitative analysis of chemical components in Aconitum carmichaeli Debeaux extract
Compound name |
Molecular Formula |
RT (min) |
Linear range (ng/mL) |
Regression equation |
R2 |
Content (mg/g) |
benzoylmesaconine |
C31H43NO10 |
2.85 |
250-5000 |
y=11.99978x+1898.13 |
0.99148 |
3.36 |
benzoylaconitine |
C32H45NO10 |
3.24 |
250-5000 |
y=6.08927x+1241.752 |
0.99131 |
0.84 |
aconitine |
C34H47NO11 |
4.90 |
250-5000 |
y=12.72896x+1933.095 |
0.99427 |
ND |

Reviewer 2 Report (New Reviewer)
Comments and Suggestions for Authors
The authors presented work that combined docking and transcriptomic analysis of ACD effects within brain and intestinal networks, with most action being involved in the brain. The authors propose that ACD works as an antidepressant and that the diarrhea is a side effect of ACD by improving neurotransmitter dysfunction. The relevant gene differences were shown using two cell lines treated with ACD.
While the premise is really intriguing, it is hard to tell from the discussion if the purpose of the paper is to define a methodology or look at ACDs effect in the brain and intestine, which both are in general interesting. From the introduction, the focus on ACDs effect seems the most natural, and thus the discussion should focus on that more. For example, a lot of work appears to have been done with identifying hub genes, but these findings were not highlighted in the discussion.
Minor corrections/suggestions:
1. The network figures are smaller and harder to visualize, with the constraint on size, an additional supplemental figure containing a larger, clearer version would be helpful.
2. There are several places in the methods and results sections where proteome is used instead of proteins, for example line 354, you “targeted 854 druggable proteomes”?
3. A brief mention of the use of the data generated by section 4.6 would be good, from initial reading I assumed the compounds in ACD were taken from literature alone, but this section suggests otherwise.
4. Volcano plots maybe more interesting than Figure 3 A&B, it would highlight the over/under expression values more, i.e. are the 1000 genes all at log2(1.5) or are many of them much higher/lower.
Author Response
Reviewer 2.
The authors presented work that combined docking and transcriptomic analysis of ACD effects within brain and intestinal networks, with most action being involved in the brain. The authors propose that ACD works as an antidepressant and that the diarrhea is a side effect of ACD by improving neurotransmitter dysfunction. The relevant gene differences were shown using two cell lines treated with ACD.
While the premise is really intriguing, it is hard to tell from the discussion if the purpose of the paper is to define a methodology or look at ACDs effect in the brain and intestine, which both are in general interesting. From the introduction, the focus on ACDs effect seems the most natural, and thus the discussion should focus on that more. For example, a lot of work appears to have been done with identifying hub genes, but these findings were not highlighted in the discussion.
Response: Thank you for taking the time and effort to review our manuscript, and for your encouraging comments regarding our study.
Minor corrections/suggestions:
- The network figures are smaller and harder to visualize, with the constraint on size, an additional supplemental figure containing a larger, clearer version would be helpful.
Response: The analysis using the network may not be clear from the figure. Therefore, we have provided additional information on the network presented in the figure in Supplementary Tables 1-4. Based on the reviewer's comments, we have added Supplementary Figures 2 and 3 to clarify this point.
Supplementary figure 2 Supplementary figure 3
- There are several places in the methods and results sections where proteome is used instead of proteins, for example line 354, you “targeted 854 druggable proteomes”?
Response: We used the term "proteome" in the database from which we downloaded the data. Therefore, we wanted to use the exact term used in the database, but after reading the reviewer's comments, we felt that it might be confusing. Based on the reviewer's comments, we changed all relevant instances of the term ‘proteomes’ to ‘proteins’.
- A brief mention of the use of the data generated by section 4.6 would be good, from initial reading I assumed the compounds in ACD were taken from literature alone, but this section suggests otherwise.
Response: Thank you for your comments. We have gone through a validation process to ensure that the results predicted by the literature in the real world. Based on the reviewer's comments, we have added the following passage to Section 2.4, where the real extracts first appear. Although this section is regarding results, we have briefly added the concepts of our research to help readers who might be confused.
Revised manuscript: In this study, we used compounds from ACD collected from the database to predict whether they would act more specifically on either the brain or intestine. To verify whether these predictions held true in real-world situations, we performed validation experiments using ACD extracts. (Lines 128-131)
- Volcano plots maybe more interesting than Figure 3 A&B, it would highlight the over/under expression values more, i.e. are the 1000 genes all at log2(1.5) or are many of them much higher/lower.
Response: Thank you again for your pertinent comments. However, we are preparing a follow-up paper on the relationship between ACD and long-term potentiation. As the reviewer said, Volcano plots are a very powerful tool that can show expression values. However, we think that a better picture can be presented in a follow-up paper that focuses on ACD and the specific therapeutic effects of each gene rather than in this paper. We could present the Volcano plot in this paper, but there may be research ethics issues in presenting this information here.

Round 2
Reviewer 1 Report (New Reviewer)
Comments and Suggestions for Authors
Author full fill all the comments of reviewer in the present revised manuscript and believe that the revised manuscript is now suitable for publication.
This manuscript is a resubmission of an earlier submission. The following is a list of the peer review reports and author responses from that submission.
Round 1
Reviewer 1 Report
Comments and Suggestions for Authors
The manuscript aims to predict the primary organ affected by ACD and verify the predictions with transcriptomic data. The study employs an interesting research strategy and involves various bioinformatic studies, however, it should be rejected due to the following reasons.
1. Introduction and overall study design:
1) It seems abrupt to directly link ACD with the digestive system or even the intestines. According to the ‘Meridian affinity’ TCM theory as mentioned in line 40, ACD’s acting on ‘the spleen meridian’ does not necessarily mean that it will treat digestive system diseases. On one hand, traditionally, ACD is rarely used alone to directly treat digestive diseases. Regarding the connection between ACD and the digestive system, ACD can only relieve abdominal pain by warming the body. On the other hand, as for the modern pharmacological research on ACD, the authors did not provide a proper review: the only article cited (DOI: 10.3390/jcm10153429; in line 46) has nothing to do with ACD. Therefore, the introduction lacks an appropriate review on ACD and its pharmacological effects to better contextualize the study’s novelty and relevance.
2) The authors stated in the first paragraph that a significant disadvantage of aconitine is its strong toxicity, hence this article. It means that the authors are focusing more on the adverse effects of aconitine rather than its efficacy. In this case, the focus of this research should be on liver, kidneys and other organs that may be damaged, rather than solely on the organs that may be treated.
3) The authors emphasized aconitine in the first paragraph, implicating the compound as a major active component of ACD. However, in Table S9, the quantitative result of aconitine in ACD extract is “ND” (not detectable?). This makes the present study’s argument confusing.
2. Network pharmacology analysis:
1) The authors showed a large number of compound-target interactions in Table S5, involving 56 ACD compounds. It is unreasonable to directly correlate the predicted number of protein targets with the effect potency of ACD, as ACD chemical components cannot have the same abundance.
2) In fact, the authors only identified two compounds in ACD extract (Table S9). There is a huge difference in ACD chemical composition between the BATMAN-TCM database and the authors’ analysis. This makes the results of both network pharmacology and the in vitro tests inconvincible.
3) When studying the target proteins of ACD in the brain tissue, the compounds that can pass through the blood-brain barrier should be emphasized. The authors should exclude compounds that cannot pass through the blood-brain barrier.
3. Transcriptome analysis:
1) There is no clear connection between the predicted target proteins and the DEGs identified by the transcriptomic analysis. The authors only discussed on the number of DEGs without any gene name mentioned. This may suggest that the in silico prediction do not align well with the actual situation.
2) In Section 2.4.1, the DEG hit in BAN/ISN of the two cell lines were low (16.54% and 4.10%). For this low hit rate, the authors made an assumption that ACD compounds bind to the predicted proteins, and in turn lead to changes in the DEGs. This assumption lacks sufficient evidence.
4. Cell choice:
Using tumor cells for transcriptomic experiments may not be the best choice, as ACD have anti-tumor effects. It would be more appropriate to use normal cells for such experiments.
Author Response
Reviewer 1
The manuscript aims to predict the primary organ affected by ACD and verify the predictions with transcriptomic data. The study employs an interesting research strategy and involves various bioinformatic studies, however, it should be rejected due to the following reasons.
Response: Thank you for your good review. We tried to convey our intention well, but we think we did not convey our intention well. The study we proposed started from the hypothesis of whether natural products act dominantly on specific organs, and since this has not been attempted yet to our knowledge, we are starting with in vitro experiments. We agree that topics such as side effect evaluation and BBB penetration are necessary, but we think that it is limited to conduct them right away because the evidence is too insufficient. To make our intention clearer, we added the word "prediction" to the paper title to emphasize that the topic of this study was to predict organ-specific drug responses using computer models. In the future, we believe that the strategies you suggested will be important key words for applying modeling studies that mimic the human body to real-world applications.
- Introduction and overall study design:
1) It seems abrupt to directly link ACD with the digestive system or even the intestines. According to the ‘Meridian affinity’ TCM theory as mentioned in line 40, ACD’s acting on ‘the spleen meridian’ does not necessarily mean that it will treat digestive system diseases. On one hand, traditionally, ACD is rarely used alone to directly treat digestive diseases. Regarding the connection between ACD and the digestive system, ACD can only relieve abdominal pain by warming the body. On the other hand, as for the modern pharmacological research on ACD, the authors did not provide a proper review: the only article cited (DOI: 10.3390/jcm10153429; in line 46) has nothing to do with ACD. Therefore, the introduction lacks an appropriate review on ACD and its pharmacological effects to better contextualize the study’s novelty and relevance.
Response: We agree that ACD does not necessarily work on digestive diseases because it acts on the "spleen meridian". However, ACD may be the key to treating digestive diseases. For example, in the classical literature called Shang Han Lun, in the case of gwol-eum disease(厥陰病) with diarrhea and abdominal pain as its main symptoms, ACD is used as the main medication. Although ACD alone may not treat digestive diseases, it may play the most important role in treating them. Also, the reference cited is not regarding ACD, but about the correlation between irritable bowel syndrome and the brain. As mentioned in the following sentence, the effect of ACD on diarrhea has not been completely verified; therefore, this reference has been cited in the context of the need for further studies on ACD action. Therefore, we think that this reference is appropriate here. We have modified the words to properly express the intention in the revised manuscript.
Lines 43–46: Treatment based on ‘meridian affinity’ has been empirically proposed and used for treating patients; however, it does not provide an accurate range of action of data-based drug therapy. Therefore, a method is necessary to address this issue.
2) The authors stated in the first paragraph that a significant disadvantage of aconitine is its strong toxicity, hence this article. It means that the authors are focusing more on the adverse effects of aconitine rather than its efficacy. In this case, the focus of this research should be on liver, kidneys and other organs that may be damaged, rather than solely on the organs that may be treated.
Response: We have mentioned the aim of this study in the last paragraph of the introduction as, “In this study, we explored the dominant pharmacological actions and therapeutic effects of ACD using various bioinformatic analyses.” If the aim is studying the side effects of ACD, it should naturally focus on the liver or kidneys. However, we tried to identify the organ that acts dominantly, and therefore, the aim of this study is not significantly related to the liver or kidneys. The topic related to the side effects has been introduced to emphasize on the need of tools for organ-specific analysis of the adverse effects, which has not yet been explored. However, since we think the reason cited by the reviewer is reasonable, we have excluded the description of side effects from the Introduction and added the related content to the Discussion of the revised manuscript.
Lines 33–35: Although ACD has been studied for its effects on several diseases and various conditions, to our knowledge, no study has been conducted for predicting the organs that are dominantly affected.
Lines 284–290: Second, the number of organs that can be studied is limited owing to the difficulty in conducting large-scale transcriptomic verification. Since ACD is highly toxic, side effects may occur. The liver and kidneys are the most important organs for evaluating drug toxicity. Because this study focused on pharmacological action, the scope was limited to the gut–brain axis. Therefore, further studies are necessary to expand the model to the liver and kidneys and assess possible side effects of the drug.
3) The authors emphasized aconitine in the first paragraph, implicating the compound as a major active component of ACD. However, in Table S9, the quantitative result of aconitine in ACD extract is “ND” (not detectable?). This makes the present study’s argument confusing.
Response: We agree with the reviewer's opinion. In fact, when ACD is distributed, it undergoes pretreatment for removing highly toxic aconitine, so aconitine is destroyed. Nevertheless, aconitine-type diterpene alkaloids, such as benzoylmesaconine and benzoylaconitine, still remain. To eliminate the readers’ confusion, aconitine has been corrected to aconitine-type diterpene alkaloids, and references regarding the efficacy and side effects of aconitine-type diterpene alkaloids have been added.
Lines 30–31: The main compounds of ACD are aconitine-type diterpene alkaloids that act as cardiotonic agents.
- Network pharmacology analysis:
1) The authors showed a large number of compound-target interactions in Table S5, involving 56 ACD compounds. It is unreasonable to directly correlate the predicted number of protein targets with the effect potency of ACD, as ACD chemical components cannot have the same abundance.
Response: The chemical components of ACD cannot have the same abundance. However, a method to analyze the efficacy of a natural product according to the amount of its constituents has not yet been developed. Therefore, the most reasonable method is to analyze the number of predicted protein targets. Whether a large number of targets is effective has not been clearly revealed; however, they are relatively more likely to show some effects than the untargeted proteins. In future research, we hope to develop a method for analyzing and predicting the efficacy of a natural product according to the amount of each constituent.
2) In fact, the authors only identified two compounds in ACD extract (Table S9). There is a huge difference in ACD chemical composition between the BATMAN-TCM database and the authors’ analysis. This makes the results of both network pharmacology and the in vitro tests inconvincible.
Response: The ACD compound information in the BATMAN-TCM database is known to provide all known ACD compounds, including those in the Chinese Pharmacopoeia. Moreover, the identification of ACD components is the result of identifying whether the herbal medicine we used is ACD; therefore, we cannot check the entire compound library. Therefore, confirming such differences in chemical composition, as observed in this study, is difficult. Network pharmacology analysis was conducted to assess whether the pharmacological action points are more distributed in the brain, starting with the known ACD compounds, and transcriptomic analysis was conducted to examine whether the on-target increases when the drug acts on brain-based cell lines.
3) When studying the target proteins of ACD in the brain tissue, the compounds that can pass through the blood-brain barrier should be emphasized. The authors should exclude compounds that cannot pass through the blood-brain barrier.
Response: Consider the blood–brain barrier (BBB) is extremely important when evaluating the pharmacodynamic properties of drugs in the brain. However, the BBB was not considered in this study. This is because the method of considering BBB in natural product compounds is still too naïve. There are two methods for evaluating BBB using natural products: (i) Considering only very small compounds and (ii) using BBB penetration data provided by TCMSP. The first method considers very small compounds, so their utility as drugs may be low, and the second method is difficult to use because it does not provide an accurate research method. In addition, GLUT expressed in the BBB recognizes glucose molecules and facilitates their entry into cells; however, if this is considered, numerous glycosides included in ACD should also be included. Considering the role of BBB in evaluating drug efficacy is a natural strategy; however, we think that at present, the BBB cannot be considered in analyzing natural products. We have added this content to Discussion.
Lines 281–284: Our presented model is at an early stage and not yet suitable for application in clinical practice. Therefore, generating a model that can consider the blood–brain barrier and ADMET (Absorption, Distribution, Metabolism, Excretion, Toxicity) is necessary in further study
Reference: Shen, Shuo, et al. "Development of GLUT1-targeting alkyl glucoside-modified dihydroartemisinin liposomes for cancer therapy." Nanoscale 12.42 (2020): 21901-21912.
- Transcriptome analysis:
1) There is no clear connection between the predicted target proteins and the DEGs identified by the transcriptomic analysis. The authors only discussed on the number of DEGs without any gene name mentioned. This may suggest that the in silico prediction do not align well with the actual situation.
Response: The transcriptomic data and information on DEGs have been uploaded to the GEO database, and accession numbers were obtained (GSE273185). The gene names are not separately mentioned because they are duplicate data reproduced with the data provided by the GEO database. However, if you think it is necessary, we can submit this information as a supplementary material. In addition, the predicted target proteins and identified DEGs were not very much related. Target proteins based on docking and network pharmacology interact directly upstream, and transcriptome analysis does not show a very high correlation because it reflects the endpoint of transcription in the nucleus. However, the use of these two research methods can be complementary to each other in increasing the reproducibility of the model, which has been described in the third paragraph of the introduction.
Data available URL: https://www.ncbi.nlm.nih.gov/geo/query/acc.cgi?acc=GSE273185
2) In Section 2.4.1, the DEG hit in BAN/ISN of the two cell lines were low (16.54% and 4.10%). For this low hit rate, the authors made an assumption that ACD compounds bind to the predicted proteins, and in turn lead to changes in the DEGs. This assumption lacks sufficient evidence.
Response: In molecular biology, it is a natural paradigm that compounds bind to proteins, and the resulting signals cause changes in the transcriptome and proteome. However, since both docking and transcriptome analyses are not perfect, examining both analyses together can provide a model that better reflects the real-world scenario for an advanced analysis. In addition, the hit rate of DEGs was not mentioned as low in the text. When comparing network pharmacology or docking analysis, 16.54% showed a high hit rate. The above analysis requires a relative comparison because the hit rate can vary depending on the model, and drawing any conclusion based on the data alone, other than the fact that BSN has a higher proportion of DEGs than ISN, is difficult.
- Cell choice:
Using tumor cells for transcriptomic experiments may not be the best choice, as ACD have anti-tumor effects. It would be more appropriate to use normal cells for such experiments.
Response: The purpose of this study is to find a dominant organ. Of course, we agree that normal cells can be used as well. However, considering that ACD is mainly used in pathological situations, it seems that tumor cells can be used without any problem.

Reviewer 2 Report
Comments and Suggestions for Authors
The research team utilized a variety of bioinformatics analysis methods, including network pharmacology, molecular docking analysis, and transcriptome analysis, to predict and verify the main organs targeted by ACD. The study found that ACD tends to act more on the brain than on the intestine and explored its potential pharmacological effects and mechanisms. The workload of this paper is moderate and has a certain degree of innovation, but the following issues need to be addressed:
1. The clarity of Figures 1 and 2 is insufficient, and there is a lack of sufficient annotation information, making the content difficult to understand.
2. The experimental section should provide detailed information on the conditions of drug treatment and the specific steps of cell culture.
3. In Sections 2.2 and 2.3, the statistical methods used in network pharmacology and molecular docking analysis need to be further refined. For example, describe the detailed process and parameter settings of the permutation test.
4. In the Materials and Methods section, it is necessary to explain in more detail the basis for the selected drug doses (low, medium, high), as well as a comparison of these doses with clinical relevance or previous studies.
5. Given that ACD contains highly toxic alkaloids, the paper needs to discuss in more detail its safety and potential toxicity. It is recommended to include a safety assessment of the dose selection and possible toxicity management strategies.
6. The discussion section needs to delve deeper into the biological significance of the results and their potential clinical applications. In addition, the limitations of the study and the direction of future research should be discussed.
Author Response
Reviewer 2
The research team utilized a variety of bioinformatics analysis methods, including network pharmacology, molecular docking analysis, and transcriptome analysis, to predict and verify the main organs targeted by ACD. The study found that ACD tends to act more on the brain than on the intestine and explored its potential pharmacological effects and mechanisms. The workload of this paper is moderate and has a certain degree of innovation, but the following issues need to be addressed:
Response: Thank you for your crucial review and important comments.
- The clarity of Figures 1 and 2 is insufficient, and there is a lack of sufficient annotation information, making the content difficult to understand.
Response: Figures 1 and 2 are diagrams that show how many action points exist in the network and how they are centrally located. They may seem unclear owing to the nature of the network; however, I think that they are relatively clear compared to those of other network analyses. To make our expression clear, we have added the number of action points distributed inside and outside the hub boundary.
Figure 1:
Figure 2:
- The experimental section should provide detailed information on the conditions of drug treatment and the specific steps of cell culture.
Response: Sections 4.4.3 and 4.4.4 already describe in detail the drug treatment conditions and details of cell culture.
- In Sections 2.2 and 2.3, the statistical methods used in network pharmacology and molecular docking analysis need to be further refined. For example, describe the detailed process and parameter settings of the permutation test.
Response: The detailed analysis method has already been described in Sections 4.2 and 4.3. To explain the method in more detail, we have added the detailed method for permutation tests in Section 4.2 of the revised manuscript.
Lines 335–339: Permutation test randomly selects the same number of nodes as the projected NP-based interacting proteins in the network and checks the ratio of hub proteins among the selected nodes. After 100,000 repetitions, the rank of actual NP hub protein ratio among the virtual hub protein ratio is assessed to calculate the p-value.
- In the Materials and Methods section, it is necessary to explain in more detail the basis for the selected drug doses (low, medium, high), as well as a comparison of these doses with clinical relevance or previous studies.
Response: We use standard operating procedures (SOPs) internally to determine capacity. We have written a manuscript on the data and SOPs that we use; however, it is currently under review. Therefore, it cannot be cited as a reference right away. Instead, we have added the used SOP used in the revised manuscript.
Lines 410–417: To assess the appropriate drug concentrations for treatment, we performed cytotoxicity tests to determine drug doses that maintained 80% cell viability (IC20). These doses were then adopted as the maximal doses for treatment and RNA sequencing (RNA-seq) analysis. For drugs with undetermined IC20, the highest treatment concentration was set to 500 µg/mL for extracts, considering both their solubility and relevance for clinical application. To confirm the influence of concentration, cells were treated with drugs at three different concentrations using 1/5 serial dilutions, thereby exposing them to low, medium, and high drug doses.
Supplementary figure: in-warehouse SOP Guidelines
- Given that ACD contains highly toxic alkaloids, the paper needs to discuss in more detail its safety and potential toxicity. It is recommended to include a safety assessment of the dose selection and possible toxicity management strategies.
Response: As answered in question 4, we have confirmed the toxicity through IC20s experiment. We think the safety assessment of dose selection and toxicity management strategy can be sufficiently handled with the SOP we are using.
- The discussion section needs to delve deeper into the biological significance of the results and their potential clinical applications. In addition, the limitations of the study and the direction of future research should be discussed.
Response: Thank you for your comments. We think that this study could focus more on biological significance. However, this model is still in its early stage, and several obstacles need to be overcome before its clinical application. Therefore, we are cautious about adding this to the manuscript. We have added future research directions on biological significance and potential clinical applications to the discussion in the revised manuscript.
Lines 281–284: Our presented model is at an early stage and not yet suitable for application in clinical practice. Therefore, generating a model that can consider the blood–brain barrier and ADMET (Absorption, Distribution, Metabolism, Excretion, Toxicity) is necessary.
Lines 297–299: In particular, the method for predicting the pharmacological mechanism of action on organs using an in-silico model may be clinically significant as it can considerably save time and cost and play an important role in ensuring drug safety.

Reviewer 3 Report
Comments and Suggestions for Authors
In this work, the authors report a bioinformatics approach for finding affected targets in human organs for the compounds from Aconitum carmichaeli and explain its application in medicine. Using the networks approach, models of brain and intestine systems were constructed, and corresponding protein targets were assigned. The authors further carried out molecular docking to identify the possible targets. It was predicted that Aconitum carmichaeli would have a more dominant effect on the brain than the intestine. After the gene enrichment, the authors chose the calcium signalling pathway as the most affected. They also suggest that ACD influences LTP by modulating calcium signalling, indicating its potential as a promising candidate for treating Alzheimer's disease.
This study should be improved by further analysis of the selected compounds. If the authors aim for the brain as a more affected organ, the blood-brain barrier and ADMET calculations should be carried out. It would be interesting to see the difference.
Also, other medical conditions should be addressed, based on the results from gene enrichment.
Author Response
Reviewer 3
In this work, the authors report a bioinformatics approach for finding affected targets in human organs for the compounds from Aconitum carmichaeli and explain its application in medicine. Using the networks approach, models of brain and intestine systems were constructed, and corresponding protein targets were assigned. The authors further carried out molecular docking to identify the possible targets. It was predicted that Aconitum carmichaeli would have a more dominant effect on the brain than the intestine. After the gene enrichment, the authors chose the calcium signalling pathway as the most affected. They also suggest that ACD influences LTP by modulating calcium signalling, indicating its potential as a promising candidate for treating Alzheimer's disease.
Response: Thank you for your comments. We hope that it will help us in improving our manuscript.
This study should be improved by further analysis of the selected compounds. If the authors aim for the brain as a more affected organ, the blood-brain barrier and ADMET calculations should be carried out. It would be interesting to see the difference.
Response: We think that extending this study by considering BBB and ADMET is important. However, this study aimed to identify the organ where natural products act more dominantly, utilizing an in vitro model. Therefore, I think that considering BBB and ADMET is beyond the scope of this work. In addition, modeling BBB and ADMET has several obstacles. If we consider potential clinical applications later, we should solve them; however, for now, applying those determinants to the system is difficult as the model is naive. In the discussion part, we have added a comment that BBB and ADMET should be considered in future studies.
Lines 281–284: Our presented model is at an early stage and not yet suitable for application in clinical practice. Therefore, generating a model that can consider the blood–brain barrier and ADMET (Absorption, Distribution, Metabolism, Excretion, Toxicity) is necessary in further study
Also, other medical conditions should be addressed, based on the results from gene enrichment.
Response: In this study, based on gene enrichment analysis, we confirmed that long-term potentiation could be improved in both cell lines. Since we wanted to focus on the gut and brain, we paid relatively less attention to other terms. We are planning to conduct follow-up studies to assess whether ACD improves memory; therefore, we also emphasized long-term potentiation. We will consider other medical conditions in future follow-up studies.

Round 2
Reviewer 1 Report
Comments and Suggestions for Authors
I would maintain my previous comments on this work that it should not be published at the present stage. Although the authors gave long explanations and modified some expressions in the manuscript, no substantive changes have been made on the major problems including the following:
1) It is meaningless to predict the drug's binding locations without considering whether the compounds can penetrate the blood-brain barrier. The transcriptomic results obtained from tumor cells SW1783 and HT29 are also not entirely appropriate.
2) Furthermore, transcriptome sequencing is typically used to validate the accuracy of the bioinformatic predictions. The article should have conducted separate GO enrichment analyses for DEGs and DP to compare the differences between the predicted and actual outcomes. However, the authors did not analyze the reliability of the predicted results and simply combined the analyses of DEGs and DP. Figure 4 indicates significant differences between DEGs and DP.
3) The "gut-brain axis" mentioned in the authors’ response is not supported by background information or the present research data. The study lacks relevance on the correlation between brain and digestive disorders.
Reviewer 2 Report
Comments and Suggestions for Authors
The authors have well addressed all my issues.
Reviewer 3 Report
Comments and Suggestions for Authors
The author's response does not meet the conditions for further consideration. The suggestions from the previous round were not properly addressed.